# Health Care Costs and Savings Associated with Increased Dairy Consumption among Adults in the United States

**DOI:** 10.3390/nu12010233

**Published:** 2020-01-16

**Authors:** Carolyn G. Scrafford, Xiaoyu Bi, Jasjit K. Multani, Mary M. Murphy, Jordana K. Schmier, Leila M. Barraj

**Affiliations:** 1Center for Chemical Regulation and Food Safety, Exponent, Inc., Washington, DC 20036, USA; xbi@exponent.com (X.B.); mmurphy@exponent.com (M.M.M.); Lbarraj@exponent.com (L.M.B.); 2IQVIA, Health Economics and Outcomes Research, Falls Church, VA 22042, USA; 3Health Sciences, Exponent, Inc., Alexandria, VA 22314, USA; Jschmier@exponent.com

**Keywords:** dairy products, chronic health outcomes, nutrition economics, costs and cost analysis

## Abstract

Background: The purpose of this study is to estimate the impact on health care costs if United States (US) adults increased their dairy consumption to meet Dietary Guidelines for Americans (DGA) recommendations. Methods: Risk estimates from recent meta-analyses quantifying the association between dairy consumption and health outcomes were combined with the increase in dairy consumption under two scenarios where population mean dairy intakes from the 2015–2016 What We Eat in America were increased to meet the DGA recommendations: (1) according to proportions by type as specified in US Department of Agriculture Food Intake Patterns and (2) assuming the consumption of a single dairy type. The resulting change in risk was combined with published data on annual health care costs to estimate impact on costs. Health care costs were adjusted to account for potential double counting due to overlapping comorbidities of the health outcomes included. Results: Total dairy consumption among adults in the US was 1.49 cup-equivalents per day (c-eq/day), requiring an increase of 1.51 c-eq/day to meet the DGA recommendation. Annual cost savings of $12.5 billion (B) (range of $2.0B to $25.6B) were estimated based on total dairy consumption resulting from a reduction in stroke, hypertension, type 2 diabetes, and colorectal cancer and an increased risk of Parkinson’s disease and prostate cancer. Similar annual cost savings were estimated for an increase in low-fat dairy consumption ($14.1B; range of $0.8B to $27.9B). Among dairy sub-types, an increase of approximately 0.5 c-eq/day of yogurt consumption alone to help meet the DGA recommendations resulted in the highest annual cost savings of $32.5B (range of $16.5B to $52.8B), mostly driven by a reduction in type 2 diabetes. Conclusions: Adoption of a dietary pattern with increased dairy consumption among adults in the US to meet DGA recommendations has the potential to provide billions of dollars in savings.

## 1. Introduction

The 2015–2020 Dietary Guidelines for Americans (DGA) encourages all Americans to consume 3 cup-equivalents (c-eq) of dairy products daily, with most choices fat-free or low-fat, to ensure adequate intakes of key nutrients including calcium, potassium, protein and vitamins A and D provided primarily by dairy [1]. As noted in the 2015 Dietary Guidelines Advisory Committee (DGAC) report [2], research also shows that the consumption of dairy products at levels consistent with dietary guidance is associated with reduced risks for many chronic health conditions, including but not limited to obesity, cardiovascular disease, type 2 diabetes, and metabolic syndrome. The majority of Americans, however, fall short of meeting recommended intakes of dairy products. Only 14% of Americans one year and older consume the recommended intake of dairy [1].

Diet is recognized to play an important role in public health with increasing epidemiological and clinical evidence supporting associations between specific foods and nutrients in maintaining and supporting good health as well as contributing to the development or prevention of disease. Health economic evaluation methods can be applied to estimate the impact of diet on the costs of chronic disease, just as these methods are used to evaluate pharmaceutical interventions. For example, health care savings associated with improved diet quality among adults in the United States (US), as measured by the Healthy Eating Index-2015, were estimated to exceed $38 billion based on estimated reductions in the disease burden for cardiovascular disease, type 2 diabetes and cancer [3]. More recently, a population attributable risk analysis reported that $2.1 billion in direct health care expenditures were attributable to low dairy consumption and the corresponding associations with chronic diseases including obesity, type 2 diabetes, heart disease, hypertension, and osteoporosis in an Australian population [4].

There is considerable evidence supporting favorable associations between dairy consumption and health outcomes. The 2010 DGAC concluded that there is moderate evidence of an inverse association between dairy products and cardiovascular disease [5]. Additional studies, including high-quality meta-analyses with findings of significant protective associations between dairy intake and health outcomes in adults [6,7,8,9,10,11], have increased the evidence base on the association between dairy products and cardiovascular disease as well as on the potential increased risk of adverse outcomes including prostate cancer [12,13] and Parkinson’s disease [14] that should be considered to fully understand the net impact of increased dairy consumption on health care costs. A 2016 systematic review concluded that there was moderate- to high-quality evidence to support favorable or neutral associations of dairy consumption with chronic health outcomes of stroke, coronary heart disease, hypertension, and type 2 diabetes [15].

The objective of this study was to quantify the net annual costs, in terms of direct medical costs (i.e., medical encounters, procedures and prescription medications) and indirect costs (mortality and lost productivity) associated with increased intake of dairy products in the US if all adults (20 years and older) were to meet the DGA recommendation of 3 c-eq/day of dairy products [1]. Both favorable and adverse associations between the consumption of dairy (total dairy, milk, cheese, and yogurt) and health outcomes were included based on a comprehensive review of the current scientific literature. Further, given the DGA’s recommendation to meet these guidelines with low-fat dairy options, where data were available, the associations between health outcomes and low-fat and high-fat dairy were examined in the model.

## 2. Materials and Methods

### 2.1. Overview

The approach and model used in this health economics evaluation to estimate the net change in health care costs associated with increased dairy consumption are summarized in Figure 1. Data inputs included (a) relative risk (RR) estimates of the association between dairy consumption and health outcomes, (b) dairy consumption among adults in the US, and (c) direct and indirect costs associated with health outcomes identified in (a) above. This study is exempt from International Review Board (IRB) approval as it is a secondary analysis of published data.

### 2.2. Dairy Consumption and Health Outcomes

#### 2.2.1. Identification of Health Outcomes

A review of the published literature using PubMed identified health outcomes associated with dairy consumption. The following search terms were used: (“dairy products”[All Fields] OR “dairy”[All Fields]) AND ((“health”[MeSH Terms] OR “health”[All Fields]) OR “chronic disease”[All Fields]) AND ((meta-analysis[ptyp] OR observational study[ptyp] OR systematic[sb] OR clinical trial[ptyp]) with limits only for human studies published in English. The aim of the literature search was to identify moderate- to high-quality studies for each outcome rather than to conduct a formal assessment of the evidence. Search results were independently screened by two authors (CGS and MMM) to determine whether each identified paper met the following inclusion criteria: (1) meta-analysis published in the past 13 years of prospective cohort studies, (2) conducted in a cohort of healthy adults (18 years and older) at risk for chronic disease, and (3) provided quantitative measures of the association between dairy products and health outcomes. Dairy products were defined to include total dairy, low-fat dairy, high-fat dairy, milk, cheese, and yogurt. Studies reporting intermediate markers of disease were excluded. No a priori selection of health outcomes was included in the search protocol; all outcomes, including adverse or favorable, were eligible for inclusion in the review. A flow chart of the study selection is provided in Figure 2. As part of the initial search and update, 38 full-text studies were reviewed and 19 were excluded due to not containing quantitative measures for the association between dairy and health outcomes or being conducted in a diseased population. The 19 meta-analyses that met the inclusion criteria were further evaluated using the Meta-analysis of Observational Studies in Epidemiology (MOOSE) checklist [16] to assess quality of reporting. All studies were of moderate or high quality, defined as a MOOSE score of >60%, and thus eligible for inclusion in the health care costs model. If more than one meta-analysis met all inclusion criteria for a given association, the most recent analysis was selected for inclusion in the health care costs model. Further, given the uncertainty inherent in combining risk estimates from individual studies based on the highest and lowest quantiles of conformance, a linear dose response analysis was preferential if both the dose response along with a high versus low comparison was available. Of the 19 studies, eight [10,12,17,18,19,20,21,22] were excluded for reasons including either earlier publications and/or with fewer studies included, limited to a high versus low comparison of intake or reported neutral associations between a dairy product and the health outcome. A summary of the eleven studies [6,7,8,9,11,14,23,24,25,26] included in the current model is provided in Table 1.

Based on our review of the evidence on the beneficial effects of dairy consumption on chronic disease outcomes, inverse associations between dairy consumption and heart disease, type 2 diabetes, colorectal cancer, and hip fractures were identified. A review of the literature on the associations between dairy intake and coronary heart disease resulted in largely null findings in dose–response analyses as well as high versus low comparisons, which is consistent with a recent systematic review on the strength of evidence of dairy products and chronic health outcomes [15]. Based on this review, coronary heart disease was not included as a health outcome in the current model. Stroke and hypertension were found to be favorably associated with total dairy, high-fat dairy (hypertension only), low-fat dairy, and milk consumption in dose–response meta-analyses [6,8,23,24]. Apart from colorectal cancer, there was limited evidence on the beneficial association between dairy consumption and cancer endpoints. The World Cancer Research Fund (WCRF) International’s Continuous Update Project (CUP) reports that “[D]iets high in calcium” were associated with a “probable decreased risk” of colorectal cancer based on studies from milk and supplements [27]. In a recent meta-analysis to update this research, researchers reported a dose–response protective association between both total dairy and milk consumption and colorectal cancer per 400 g/day and 200 g/day of intake, respectively [11]. Further, an additional meta-analysis published after the WCRF publication reported protective associations with total dairy as well as high- and low-fat dairy products [25]. Two recent meta-analyses reported on the dose–response association between dairy consumption and type 2 diabetes [7,9]. In both meta-analyses, an increase in 200 g/day of total dairy consumption was associated with a 3% reduction in the incidence of type 2 diabetes and a borderline inverse dose–response association between low-fat dairy consumption. High-fat dairy was shown to have a neutral association in both meta-analyses along with milk as reported by Gijsbers et al. [7]. The association between yogurt and type 2 diabetes was associated with a 6% reduction in type 2 diabetes for each 50 g/day of consumption [7]. In a meta-analysis by Bian et al. [26], the association between dairy consumption and risk of hip fracture was summarized based on nine cohort studies and seven case-control studies. In the analyses limited to cohort studies only, yogurt and cheese were associated with a significant reduced risk of hip fractures based on four data points each when comparing high versus low categories of intake and therefore were included in the current model. Milk was not associated with reduced risk of hip fractures in either the dose–response analysis or the high versus low analysis.

Increased risks of prostate cancer and Parkinson’s disease with dairy consumption were identified. Specifically, in 2018, the WCRF CUP concluded that there was limited evidence showing that high dairy consumption increases the risk of prostate cancer [27]. This finding is based on a meta-analysis that found statistically significant increased risk of total prostate cancer in a dose–response association with total dairy, milk, and cheese [13]. The potential for an increased risk for Parkinson’s disease from the consumption of dairy products was based on a 2014 meta-analysis of seven studies [14]. The meta-analysis relied upon in the Parkinson’s disease review includes seven studies on total dairy and four studies on milk. The association between dairy and Parkinson’s disease is thought to be due, in part, to contaminants, such as pesticides, in milk. However, a recent review supports the possibility that there may be a biological mechanism associated with the urate-lowering effects of dairy products [28]. Several recent reviews note the limited evidence and speculative hypotheses associated with dairy consumption and prostate cancer and Parkinson’s disease risk [29,30]. While recognizing the potential for this adverse association, due to the limited evidence, these two outcomes were excluded as part of a secondary analysis in the present study.

Summary relative risk (RR) measures were extracted for the selected health outcomes associated with dairy consumption including a reduced risk of stroke, hypertension, type 2 diabetes, colorectal cancer, and hip fractures and an increased risk of Parkinson’s disease (Table 2). All associations included in the current model were based on statistically significant linear dose–response risk estimates with the exception of the associations between total dairy and Parkinson’s disease and cheese and yogurt and hip fractures where the summary risk estimates quantified the comparison between high and low categories of intake.

#### 2.2.2. Costs Associated with Health Outcomes

Annual direct medical costs as well as indirect costs for the selected health outcomes were based on a review of recent literature using data from the American Heart Association (2014–2015) [29], the American Diabetes Association (2017) [30], the National Cancer Institute (2010) [31], and reports in the published literature for Parkinson’s disease [32] and represent both health care payer and societal perspectives; all costs were inflated to 2018 US dollars [33] (Table 3). An important challenge in estimating net changes in costs is that chronic health outcomes such as heart disease and type 2 diabetes have similar risk factors, which are likely to play a role in mediation or interaction along the proposed causal pathways. For example, type 2 diabetes and hypertension are both established risk factors for heart disease. To address these issues, costs for one health outcome that may include the cost of co-morbidities and/or risk factors of another health outcome were adjusted to reflect this overlap attributed to each outcome to the extent the data allowed. In the 2019 American Heart Association’s update, the costs for hypertension do not include the costs for heart disease and reflect the costs of hypertension alone. In cost estimates for type 2 diabetes, the proportion of costs attributed to cardiovascular complications was excluded to estimate the net annual costs for type 2 diabetes alone [30,34] (see Table 3). A 2017 study noted that non-cancer causes of death were high in patients with colorectal and prostate cancer, with many of these patients dying from heart disease [35]. Therefore, when estimating the costs for the last year of life for colorectal and prostate cancer, the proportion of the patients who die from other causes was used to adjust the net annual costs for colorectal and prostate cancer. The use of these adjustments allowed for consideration of the isolated costs for hypertension, type 2 diabetes, and colorectal and prostate cancer without including the costs from heart disease and minimized the extent of double counting that could result from the overlap of risk factors and co-morbidities.

#### 2.2.3. Dairy Consumption among Adults in the US

Estimates of dairy consumption among adults in the US were based on food consumption records collected in the What We Eat in America (WWEIA) component of the National Health and Nutrition Examination Survey (NHANES) conducted in 2015–2016 (WWEIA, NHANES 2015–2016) [41], and the Food Patterns Equivalents Database (FPED) 2015–2016 [42] developed by the US Department of Agriculture that translates each food into food components. The NHANES datasets provide nationally representative nutrition and health data to estimate nutrition and health status measures in the US. The sample for this analysis was limited to adults in the US 20 years of age or older with reliable dietary recalls as defined by the National Center for Health Statistics on Day 1 of the data collection (*n* = 5017). All analyses to estimate the consumption of dairy products were conducted using Stata^®^ Version 12.1. Mean total dairy intake (i.e., milk, cheese, and yogurt) among the US adult population was estimated to be 1.49 c-eq/day (Table 4). Therefore, the increase in dairy consumption that would be required for the US adult population to meet the 3 c-eq/day DGA recommendation would be an additional 1.51 c-eq/day of dairy products. The majority of dairy consumption was from milk (0.63 c-eq/day) and cheese (0.73 c-eq/day) while yogurt contributed minimally to the total intake (0.09 c-eq/day). The remaining 0.04 c-eq/day was from the consumption of whey (not included in current analysis). Total dairy, milk, and cheese consumption among men only was also estimated for incorporation into the analysis for prostate cancer (Table 4).

#### 2.2.4. Model Structure

Summary RR estimates and corresponding lower and upper 95% confidence intervals quantifying the association between increased dairy consumption and health outcomes were combined with the increase in consumption required to meet the dairy recommendation; the resulting change in risk was used to estimate the impact on costs. Total annual costs were reduced or increased proportionally to reflect the change in risk of each health outcome with the change in costs estimated using the equation below:$$\Delta Cost_{i}=\left[\frac{RR_{i,\;adj}-1\;}{\Delta DC_{cited}}\;x\;\left(I_{i}+D_{i}\right)\;x\;\Delta DC_{US\;adults}\right]$$
where:Δ*Cost**_i_* = total annual change in costs for selected health outcome;*i* = index for selected health outcome (e.g., type 2 diabetes);*RR**_i, adj_* = adjusted RR for health outcome (*i*) (Table 2);Δ*DC**_cited_* = change in dairy consumption (g/day) associated with RR for health outcome (*i*) (Table 2);*I**_i_* = annual indirect costs associated with health outcome *(i)* (Table 3);*D**_i_* = annual direct costs associated with health outcome *(i)* (Table 3);Δ*DC**_US adults_* = change in dairy consumption (g/day) to meet DGA recommendation of 3 c-eq/day (Table 4).


The RRs associated with each of the selected health outcomes are provided in Table 2. The relationship between dairy consumption and risk of disease was assumed to be linear. RR measures associated with total dairy, cheese, and yogurt were reported or a “high” versus “low” comparison with the majority of the original studies reporting RRs based on quintiles of intake. Therefore, to relate dairy consumption estimates, as reported in meta-analyses, to the consumption estimate among the adult US population, the 10th and 90th percentiles (i.e., the medians of the lower and upper quintiles) of the consumption distributions were estimated. For total dairy, the 10th and 90th percentiles were estimated to be 10 g and 541 g, respectively, and therefore, the RR associated with a high versus low total dairy intake was estimated to be associated with an increased consumption of 531 g total dairy. Similarly, the 90th percentiles of cheese and yogurt consumption were 133 and 100 g, respectively, while the 10th percentiles were zero grams in both cases and therefore, the corresponding high versus low RR was associated with an increased consumption of 133 g cheese and 100 g yogurt.

Two sets of analyses were conducted. The primary analysis was based on all beneficial and adverse associations between dairy consumption and health outcomes as noted in Table 2 (See bold font). A secondary analysis was conducted that included only beneficial associations.

Within each of the two sets of analyses, two scenarios were developed that differed with respect to how dairy product (i.e., total, milk, cheese, yogurt) consumption among adults in the US was assumed to increase to meet the DGA’s recommendation of 3 c-eq/day. For each scenario, the same increases in values corresponding to total dairy, as shown in Table 4, were assumed for high- and low-fat dairy consumption.

*Scenario 1:* Population mean intakes of milk, cheese and yogurt were increased to proportions by type as specified in US Department of Agriculture Food Intake Patterns [2]. In these patterns, total dairy intake of 3 c-eq/day is comprised of 51% fluid milk, 45% cheese, 2.5% yogurt, and 1.5% soy milk (not included in model). Scenario 1 is intended to reflect a shift in dairy consumption that could occur if all adults in the US increased dairy consumption by 1.51 c-eq/day to meet the DGA recommendation while maintaining the same relative proportions of dairy by type.

*Scenario 2:* Intake of each type of dairy product was increased assuming the consumption of only that dairy type. Scenario 2 is an alternate modelling approach designed to illustrate the impact of a single dairy type (e.g., yogurt) on net annual costs if the entire gap in dairy consumption were to be met exclusively through the increased consumption of 1.51 c-eq per day of a single dairy type with the exception of yogurt. Given the current low mean intake of yogurt among the adult US population of 21 g/day (i.e., 0.09 c-eq/day), an increase of 1.51 c-eq is more than 15 times the current intake and results in a modelled consumption that is outside the range of intakes reported in many of the prospective studies that provided risk estimates in the selected meta-analyses. For milk and cheese, the modelled increase for Scenario 2 is approximately two times the current intake. Therefore, for yogurt only, the modelled increase in Scenario 2 was selected to be 100 g (~0.4 c-eq) or ~5 times the current intake and is equivalent to the intake of the high-end (90th percentile) consumer in the adult US population. The model was repeated three times, once each for milk, cheese, and yogurt.

Total net changes in annual costs within each dairy product type were estimated by summing costs for all relevant health outcomes after adjustment for overlapping risk factors and co-morbidities:(1)$$\Delta Total\;Cost_{DC}=\overset{n}{\underset{i=1}{\sum }}\Delta Cost_{i}$$
where:$\Delta Total\;Cost_{DC}$ = total net change in annual costs;*i* = index for selected health outcome (e.g., type 2 diabetes);*n* = number of health outcomes included for each dairy type;$\Delta Cost_{i}$ = annual change in costs associated with health outcome *i*.


The lower and upper 95% confidence intervals around the RR estimates were included to provide a potential range in costs associated with each health outcome. These lower and upper cost estimates for all health outcomes were also summed, with the caveat that the resulting range no longer reflects a 95% confidence interval but rather a range of the potential annual costs associated with that particular dairy type. Further, annual costs within a health outcome across the three dairy types (i.e., milk, cheese, and yogurt) cannot be summed to estimate the net changes for total dairy given that risk estimates for each dairy type are summary measures from different individual studies.

## 3. Results

Under Scenario 1, in which estimated dairy consumption among the US adult population was increased to meet the 3 c-eq/day DGA recommendation in proportions of milk, cheese, and yogurt consistent with USDA Food Intake Patterns, $12.5 billion (B) (range: $2.0B to $25.6B) in annual cost savings was estimated based on a modelled increase in total dairy consumption (Table 5) resulting from a reduction in risk of type 2 diabetes, hypertension, stroke, and colorectal cancer and an increased risk of Parkinson’s disease and prostate cancer. In the secondary analysis limited to beneficial outcomes, the annual cost savings was $16.1 billion (B) (range: $7.6B to $27.2B). The largest annual cost savings by dairy fat type was associated with low-fat dairy ($14.1B; range: $0.8B to $27.9B). Increased intake of 0.87 c-eq of milk and 0.54 c-eq of cheese consumption was estimated to yield $4.1B and $1.4B in cost savings each, respectively, in the primary analysis (Table 6). Increased milk consumption was associated with an increased risk of Parkinson’s disease and prostate cancer, resulting in increased annual costs of -$3.3B at the lower end of the range and increased costs savings of $4.1B and $11B at the mean and upper end of the range, respectively.

In Scenario 2, where the intake of each dairy type (i.e., milk, cheese, yogurt) was increased assuming the consumption of only a single dairy type to meet the DGA recommendation of 3 c-eq/day in the case of milk and cheese and an increase of 100 g of yogurt, the largest estimated annual cost savings was associated with increased yogurt consumption ($32.5B; range: $16.5B to $52.8B). This savings was based on a reduction of both type 2 diabetes and hip fractures and held constant in the secondary analysis due to the lack of a reported association with adverse outcomes.

## 4. Discussion

This study illustrates the significant potential health care cost savings associated with increasing dairy consumption among adults in the US to meet the current DGA recommendations. Our analysis of the 2015–2016 WWEIA, NHANES consumption data indicates that 87.1% of adults consume less than the DGA recommendation of 3 c-eq of dairy per day [41]. Thus, there is a large opportunity to increase consumption with significant impact on chronic health outcomes. Cost savings estimated in the current study are based on a detailed review of the epidemiological evidence of both adverse and favorable associations between dairy consumption and chronic health outcomes. An annual cost savings of $12.5 billion (B) (range: $2.0B to $25.6B) could be achieved if, on average, all adults in the US were to adopt a dietary pattern with an additional 1.51 c-eq/day total dairy by types in proportions consistent with USDA Food Intake Patterns. This increase in total dairy of unspecified fat type is slightly lower than the estimated $14.1B associated with increased intake of low-fat dairy, due in part to the lack of reported associations between low-fat dairy and adverse outcomes. In the alternate modelled scenario, where an individual would achieve the gap by consuming a single type of dairy product, increased milk and cheese intake could result in $6.6B and $3.4B in mean annual cost savings, respectively. Both changes in consumption patterns require an increase of 1.51 c-eq (i.e., ~1.5 cup of milk or ~3 oz of cheese), which could be regarded as a realistic change in dietary choices.

Type 2 diabetes contributed to over half of the overall cost savings from total dairy consumption. While risk reduction estimates per 200 g/day of total dairy for type 2 diabetes were similar or smaller compared to other health outcomes, the large contribution of type 2 diabetes to total savings is driven by the high direct and indirect costs associated with the condition, even after adjusting for overlapping pathways. Reduced risk of type 2 diabetes from a dietary pattern with an additional 100 g/day of yogurt accounted for the largest estimated annual cost savings ($32.5B; range: $16.5B to $52.8B) (i.e., Scenario 2). This estimate reflects the high costs associated with type 2 diabetes and a 6% reduction in risk from the consumption of 50 g/day of yogurt. This risk reduction assumes linearity, however, the meta-analysis reported that there is a levelling of benefits after 80–125 g yogurt intake [7]. While this scenario may not reflect a realistic or achievable dietary change for all adults in the US based on current dietary preferences, it illustrates the potential importance of yogurt consumption as a component of total dairy intake for the risk reduction of type 2 diabetes.

Adverse associations of dairy consumption with prostate cancer and Parkinson’s disease impacted estimated cost savings for total dairy by reducing the total estimate by $3.6B, largely due to costs associated with Parkinson’s disease ($3.4B). However, the estimated mean costs remained an overall savings of $12.5B (range: $2.0B to $25.6B). Cost savings associated with milk consumption were most impacted by the adverse associations with approximately a 50% reduction in mean estimated cost savings when including the adverse outcomes compared to the secondary analysis and a lower range that results in increased costs in both Scenario 1 and 2. There is limited evidence on the potential causal mechanism attributed to dairy consumption for both outcomes as noted in large, systematic reviews [27,29,30]. These outcomes were included for completeness in the current analysis, but their inclusion should be viewed with caution.

Previous studies have estimated the potential health care savings associated with dairy consumption [4,43]. In McCarron and Heaney’s analysis, they estimated first-year savings of approximately $26B if adult Americans increased their intake of dairy foods to 3 to 4 servings/day [43]. This estimate is approximately 1.6 times the estimated mean of $16.1B for net annual cost savings associated with total dairy consumption in the current secondary analysis and double the estimated mean of $12.5B for net annual cost savings in the primary analysis. The higher estimate from McCarron and Heaney can be explained by methodological differences including underlying assumptions and therefore, a direct comparison between the two studies is difficult. The current study conducted a detailed review of the recent scientific literature with set criteria for inclusion in the study, including requiring that dairy consumption be the exposure variable and a disease endpoint defined as an outcome with both adverse and favorable associations between dairy consumption and disease endpoints included in the assessment. The previous study presented a first year and a 5 year total projected cost savings. In the first year, health outcomes that contributed to the $26B in savings included obesity, osteoporosis, nephrolithiasis, and pregnancy outcomes along with hypertension and type 2 diabetes. We did not identify moderate to high-quality meta-analyses that reported significant associations between osteoporosis, obesity, pregnancy outcomes, or nephrolithiasis with disease endpoints and dairy consumption. We did find several meta-analyses that looked at calcium intake (diet and/or supplementation) and osteoporosis but none that reported on dietary dairy intake and therefore, these studies did not meet our inclusion criteria. Meta-analyses on obesity were either null or limited to reported changes in abdominal obesity, body weight and/or anthropometric measurements with dairy consumption, lacking the data necessary to translate these findings to a disease endpoint such as obesity [22,44]. In a more recent analysis based on the Australian population, $2.1B (2010–2011 US dollars) in direct health care expenditure was attributed to low dairy consumption [4]. This estimate is lower than the mean estimate for total dairy in the current study. Explanations for the differences include differences in medical costs for each condition between the US and Australia, the methods of adjustments made for double-counting, and the fact that the Australian analysis is based on direct health care expenditures and estimates the indirect costs using disability adjusted life-years (DALYs). The current analysis relied upon indirect cost data from the literature, however, these data were not available for colorectal cancer and therefore are not included in the cost savings estimates. However, even with the methodological and data source differences among these analyses, all studies support a significant potential for cost savings related to total dairy consumption and chronic health outcomes in the general adult population.

While the current model is based on an approach and assumptions that are consistent with other economic analyses, there are limitations and uncertainties to consider. The use of observational data to measure the association between dairy consumption and health outcomes will include the potential for residual confounding in the estimates of the RR. However, given the chronic nature of the health outcomes associated with dietary variables including dairy, randomized controlled trials are not realistic or feasible and many of those that have been published are limited to calcium supplementation or reduced fat dairy interventions and tend to be in at-risk populations (e.g., obese individuals) with insufficient follow-up for chronic disease outcomes. These characteristics of controlled trials limit the generalizability of these results to the US adult population. To capture the potential uncertainty inherent in observational studies, the lower and upper limits of the 95% confidence interval around the summary risk estimates were included to provide a range of potential cost savings Further, the definition of “total dairy” can vary significantly from study to study included in each meta-analysis. While the inclusion of milk, cheese, and yogurt in the category of total dairy is standard, some definitions also included products such as fermented dairy, ice cream and butter which increased heterogeneity among the studies included in the summary risk estimate and attenuate results observed due to misclassification and measurement bias.

Consistent with similar economic analyses, the current model assumes that the relationship between dairy consumption and risk of disease is linear; that is, as consumption increases, risk will change by a set amount, which in turn will have a linear effect on the change in health care costs. Our criterion that preferentially selects meta-analyses that evaluated the dose–response association and provided risk estimates per dose of dairy is a strength of this analysis when compared to a high versus low comparison. This refinement requires more detailed information from each study included in the meta-analysis and helps to reduce the potential misclassification bias that occurs from basing comparisons on extreme quantiles of intake from studies as it is possible that a high consumer in one study population could be classified as a low consumer in another study population. The current model’s stricter inclusion criterion should help to reduce overestimation of effects but, in turn, may both over and underestimate cost savings if the association follows a non-linear dose response and has a threshold level above or below the level at which effects occur. For example, in the association between yogurt and type 2 diabetes, the linear model estimated a 6% reduction per 50 g/day increase in yogurt consumption while the non-linear model reported a 14% risk reduction when comparing 80 g/day to 0 g/day with no further risk reduction observed above 80–125 g/day [7]. In this example, the linear model underestimates the risk reduction up to 80–125 g/day and overestimates risk reduction for yogurt intakes above ~125 g/day. It is also possible that the linear assumption between disease and costs is not valid, or not valid across the selected conditions, or that thresholds exist in the relationship between disease severity and cost. Further, there may be substantial costs associated with undiagnosed and pre- diabetes [45] and in patients with Parkinson’s disease in the year prior to diagnosis [46]. Since this model is limited to clinically diagnosed diseases, it likely underestimates costs and in turn, savings.

The current model does not account for substitution effects that may occur from increasing dairy consumption. The model assumes adoption of a dietary pattern that reflects a mean per capita increase in dairy intake of 1.51 c-eq/day to meet the 3 c-eq/day DGA recommendation across the US adult population. Each estimated change in annual net costs based on the current model assumes that an individual’s increased dairy consumption would all come from a specified dairy type (e.g., high-fat dairy, milk). Intake of milk, cheese, and yogurt in Scenario 1 assumes that incremental intakes by type result in total intake of each dairy in proportions as specified in USDA Food Intake Patterns while Scenario 2 assumes that the entire dairy gap would be met, or partially met in the case of yogurt, by consuming a particular dairy type. However, given that individuals would be increasing intake of the dairy food group, decreases in the consumption of other food groups would be necessary to maintain caloric intake. Any substitution or replacement in an individual’s diet would most likely have an effect on risk factors, disease pathways and outcomes. The relative risks in the underlying studies included in the meta-analyses selected for inclusion were adjusted for demographic, lifestyle, and dietary factors such as energy intake or intake of particular foods or dietary components (e.g., saturated fat). The resulting summary risk estimates used in the current analysis are intended to reflect dietary patterns differing in the amount of dairy consumed independent of the effects of the other measured factors. In the application of risk estimates for a specified increase in a food component, the scenario approach like that conducted in the current analysis assumes that dietary patterns in the modelled population shifts in a similar way to accommodate the increased consumption of dairy. Further, given the mean per capita shift applied to the entire population, the model does not consider the extreme ends of the dairy intake distribution and how that could affect the proportion of adults who would benefit from particular levels of intake increases. There may be adults with a daily intake just below the recommended amount who would benefit from as little as one additional serving daily, while there are also adults whose intake is so low that even an increase of 1.51 servings daily might not be sufficient to reduce the risk of dairy-associated health concerns. While the model could have considered various distributions of intake in the population, without sufficient data to inform which distributions would be appropriate, the inclusion of this element would have added precision to the model without necessarily increasing accuracy.

The model inputs are largely mean point estimates. Health care costs and consumption data are often right skewed, which indicates the mean value will be higher than another central tendency estimate such as the median or the geometric mean. Therefore, the use of the mean value for both the consumption data and health care costs data would result in an underestimate of the increase in consumption required to meet the DGA recommendation and a potential overestimate of the costs associated with each health outcomes. Further, indirect costs were not available or not distinguishable in the published literature for colorectal cancer and hip fractures, respectively.

The benefits of changes in dietary patterns are typically not immediate, and even more so when considering the chronic health outcomes in this model. Although exact timelines are unknown and may vary by condition, it is likely that there are incremental improvements over time. The current model assumes that any shift in intake has had time to result in the benefits included in the model. In fact, the opposite may be true, in that there may be health economic benefits even among undiagnosed or preclinical populations. Such may be the case for pre-diabetes, which is associated with substantial costs [45] but is not included in this model. Other studies have suggested a similar concern with post-diagnosis costs may be relevant for other conditions. In particular, among patients who have had a stroke, costs appear to be highest in the year following the initial event and moderate in subsequent years [47]. This model did not explicitly consider long-term patterns of costs, rather the costs used for each condition likely reflect a distribution of patients with disease of various duration and severity.

The current model does not consider the potential health and economic impact of side-effects relating to consuming additional dairy, particularly for lactose-intolerant individuals. However, potential strategies to cope with dairy intolerance include incorporation of dairy in small portions or selection of yogurt or cheese in place of milk. This model also did not consider the economic costs associated with increasing dairy consumption, in terms of costs for purchasing food, implementing a public health intervention, or macro-level policy implications of shifts in demand for dairy. The current study is theoretical and can be used to illustrate the potential economic impact of any public health intervention focused on increasing the US population’s adherence to the DGA recommendations for dairy consumed as a combination of milk, cheese, and yogurt. The net annual health care cost savings presented in this study would only be realized if interventions aimed at increasing dairy consumption are fully effective and if the observed associations between dairy consumption and health outcomes are true.

## 5. Conclusions

The current model identifies a potential reduction in dairy-related health care costs and illustrates that a simple, realistic dietary change at the population level consisting of adoption of a dietary pattern with increased daily dairy consumption results in an economic benefit. While there are limitations to the modelling approach and data incorporated that are consistent with other economic analyses, the results of the current study can be used to support efforts to understand the health economic implications of increased dairy consumption and the importance of helping adults in the US meet the daily recommended dairy intakes.

## Figures and Tables

**Figure 1 nutrients-12-00233-f001:**
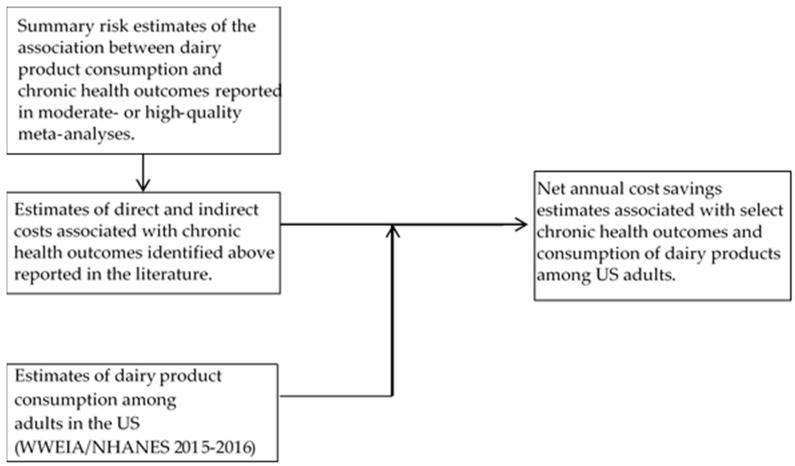
Overview of data inputs and model to estimate net annual cost savings associated with increased dairy consumption among adults in the United States (US). WWEIA = What We Eat in America; NHANES = National Health and Nutrition Examination Survey.

**Figure 2 nutrients-12-00233-f002:**
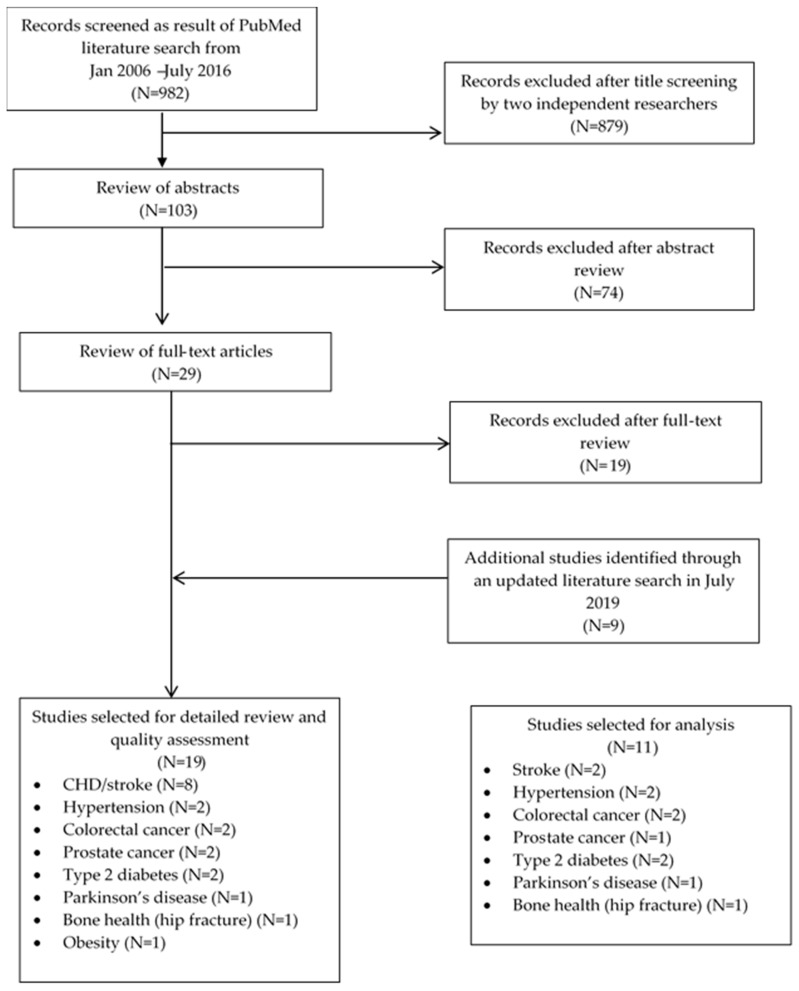
Study selection flow chart for review of published meta-analyses measuring the association between dairy consumption and chronic health outcomes.

**Table 1 nutrients-12-00233-t001:** Summary of selected studies measuring the association between dairy consumption and health outcomes.

Health Outcome(s)	Selected Study (MOOSE Rating)	Study Population	Endpoints Measured	Dairy Types	Comparator
Cardiovascular diseases and related outcomes	Bechthold et al. 2017 [23] (81%)	N = 24 studies (Europe = 15, US = 8, Asia = 1) 5.4–26 y of follow-up Prospective cohort studies, case-cohort, nested case-control, RCTs	Fatal/nonfatal coronary heart disease; stroke; heart failure	Total dairy (low-fat and high-fat)	High vs. low intake; 200 g/day
	de Goede et al. 2016 [8] (97%)	N = 18 studies (US, Europe, Nordic countries, Australia, Japan, China, Singapore); 8 to 26 years of follow-up; 762,414 individuals and 29,943 stroke events Prospective cohort studies	Total stroke and ischemic, hemorrhagic, or fatal stroke	Total dairy (low-fat and high-fat) Fermented dairy Milk (low-fat and high-fat), cheese, yogurt	Milk: 200 g/day Cheese: 40 g/day Yogurt:100 g/day
Hypertension	Schwingshackl et al. 2017 [24] (86%)	N = 9 studies (Europe = 5, US = 3, Asia = 1) 116,415 subjects 2–15 y of follow-up Prospective cohort studies, case-cohort, nested case-control, RCTs	Incidence (SBP ≥ 140 mm Hg OR DBP ≥ 90 mm hg OR anti-HT medication use)	Total dairy (low-fat and high-fat)	High vs. low intake; 200 g/day
	Soedamah-Muthu et al. 2012 [6] (72%)	N = 9 studies Prospective cohort studies Duration of follow-up: 2 to 15 y	Incidence (SBP ≥ 140 mm Hg OR DBP ≥ 90 mm hg OR anti-HT medication use)	Total dairy (low-fat and high-fat) Milk, cheese, yogurt	200 g/day
Colorectal cancer	Schwingshackl et al. 2017 [25] (89%)	N = 18 studies (Europe = 8, US = 8, Asia = 2) 1,629,366 subjects Duration of follow-up: 3.3–26 y Prospective cohort, longitudinal, follow-up, case-cohort, nested case control studies	Colorectal cancer	Total dairy (low-fat and high-fat)	High vs. low intake; 200 g/day
	Vieira et al. 2017 [11] (72%)	N = 10 studies (Europe and US) Prospective cohort studies, case-cohort, nested case-control, RCTs	Colorectal cancer	Total dairy Milk	Total dairy: 400 g/day Milk: 200 g/day
Prostate cancer	Aune et al. 2015 [13] (86%)	N = 15 studies (total dairy, milk); N = 11 studies (cheese); N = 6 studies (yogurt) Prospective cohort studies	Total prostate cancer, non-advanced, advanced, fatal	Total dairy, milk, low-fat milk, whole milk, cheese, yogurt	Total dairy: 400 g/day Milk:200 g/ day Cheese: 50 g/day Yogurt: 100 g/day
Type 2 diabetes	Schwingshackl et al. 2017 [9] (78%)	N = 21 studies (Europe N = 8, US N = 7, Asia N = 4, Australia N = 2) Prospective cohort studies, nested case-control studies, case-cohort studies	Type 2 diabetes	Total dairy (low-fat and high-fat)	Total dairy: 200 g/day
	Gijsbers et al. 2016 [7] (81%)	N = 20 articles/22 studies/23 populations (US, Europe, Asia, Australia) Prospective cohort studies Duration of follow-up: 2.6–30 y	Type 2 diabetes	Total dairy (low-fat and high-fat) Milk, cheese, yogurt	Total dairy/milk:200 g/day Cheese: 10 g/day Yogurt: 50 g/day
Parkinson’s disease	Jiang et al. 2014 [14] (72%)	N = 5 studies (US, Finland, Greece) and 7 data points; follow-up from 8.45 to 41 y Prospective cohort studies	Parkinson’s disease	Total dairy, milk	Total dairy: high vs. low intake Milk: 200 g/day
Hip fracture	Bian et al. 2018 [26] (78%)	N = 18 studies (Europe = 7, US = 5, Asia = 4, Australia = 1, Europe/Canada/Australia = 1) 381,987 subjects Prospective cohort and case control studies	Hip fracture	Total dairy Milk, yogurt, cheese, cream	High vs. low intake; Milk: 200 g/day

DBP = diastolic blood pressure; g/d = grams per day; HT = hypertension; incr. = increments; MOOSE = Meta-analyses Of Observational Studies in Epidemiology checklist; N = number; RCT = randomized controlled trial; SBP = systolic blood pressure; vs. = versus; y = years.

**Table 2 nutrients-12-00233-t002:** Summary of published risk estimates for health outcomes associated with dairy consumption.

Health Outcome	Relative Risk (95%CI)	Comparator	Source
	**Total dairy**		
**Stroke**	**0.96 (0.94, 0.98)**	**per 200 g/day**	**[23]**
**Hypertension**	**0.95 (0.94, 0.97)**	**per 200 g/day**	**[24]**
**Type 2 diabetes**	**0.97 (0.94, 0.99)**	**per 200 g/day**	**[9]**
Hip fractures	1.02 (0.93, 1.12)	High vs. low	[26]
**Colorectal cancer**	**0.93 (0.91, 0.94)**	**per 200 g/day**	**[25]**
**Parkinson’s disease**	**1.40 (1.20, 1.63)**	**High vs. low**	**[14]**
**Prostate cancer**	**1.07 (1.02, 1.12)**	**per 400 g/day**	**[13]**
	**High-fat dairy**		
Stroke	0.99 (0.97, 1.02)	per 200 g/day	[23]
**Hypertension**	**0.97 (0.93, 0.98)**	**per 200 g/day**	**[24]**
Type 2 diabetes	1.00 (0.96, 1.04)	per 200 g/day	[9]
Hip fractures	--	--	
**Colorectal cancer**	**0.91 (0.86, 0.97)**	**per 200 g/day**	**[25]**
Parkinson’s disease	--	--	
Prostate cancer	--	--	
	**Low-fat dairy**		
**Stroke**	**0.98 (0.95, 1.00)**	**per 200 g/day**	**[23]**
**Hypertension**	**0.96 (0.93, 0.99)**	**per 200 g/day**	**[24]**
**Type 2 diabetes**	**0.97 (0.94, 1.00)**	**per 200 g/day**	**[9]**
Hip fractures	--	--	
**Colorectal cancer**	**0.94 (0.88, 1.00)**	**per 200 g/day**	**[25]**
Parkinson’s disease	--	--	
Prostate cancer	--	--	
	**Milk**		
**Stroke**	**0.93 (0.88, 0.98)**	**per 200 g/day**	**[8]**
**Hypertension**	**0.96 (0.94, 0.98)**	**per 200 g/day**	**[6]**
Type 2 diabetes	0.97 (0.93, 1.02)	per 200 g/day	[7]
Hip fractures	1.00 (0.94, 1.07)	per 200 g/day	[26]
**Colorectal cancer**	**0.94 (0.92, 0.96)**	**per 200 g/day**	**[11]**
**Parkinson’s disease**	**1.17 (1.06, 1.30)**	**per 200 g/day**	**[14]**
**Prostate cancer**	**1.03 (1.00, 1.06)**	**per 200 g/day**	**[13]**
	**Cheese**		
Stroke	0.97 (0.94, 1.01)	per 40 g/day	[8]
Hypertension	1.00 (0.98, 1.03)	per 200 g/day	[6]
Type 2 diabetes	1.00 (0.99, 1.02)	per 10 g/day	[7]
**Hip fractures**	**0.68 (0.61, 0.77)**	**High vs. low**	**[26]**
Colorectal cancer	--	--	
Parkinson’s disease	1.26 (0.99, 1.60)	High vs. low	[14]
**Prostate cancer**	**1.10 (1.03, 1.18)**	**per 50 g/day**	**[13]**
	**Yogurt**		
Stroke	1.02 (0.90, 1.17)	per 100 g/day	[8]
Hypertension	0.99 (0.96, 1.01)	per 200 g/day	[6]
**Type 2 diabetes**	**0.94 (0.90, 0.97)**	**per 50 g/day**	**[7]**
**Hip fractures**	**0.75 (0.66, 0.86)**	**High vs. low**	**[26]**
Colorectal cancer	--	--	
Parkinson’s disease	0.95 (0.76, 1.20)	High vs. low	[14]
Prostate cancer	1.08 (0.93, 1.24)	per 100 g/day	[13]

Note: Bolded rows indicate statistically significant risk estimates included in the primary analyses. --, no published meta-analyses available; g = grams; vs. = versus.

**Table 3 nutrients-12-00233-t003:** Estimated annual direct and indirect health care costs (Billions $) for selected health outcomes based on published studies.

	Annual Direct and Indirect Costs (Billions $)			
Health Outcome	Direct	Indirect	Total	Assumptions and Adjustments
Stroke	30.3	18.9	49.2	Annual average cost from 2015–2016 [29].
Hypertension	55.5	5.0	60.4	Annual average cost from 2015–2016; limited to hypertension without heart disease [29].
Type 2 diabetes	207.6	105.6	313.2	Annual average cost from 2017 for total expenditures and indirect costs for diabetes ($327B) [30] and assuming 96% of diabetes cases are type 2 diabetes based on a cited prevalence of 1.25 million type 1 diabetes cases out of total prevalence of 30.3 million Americans with diabetes in 2015 [36]. The proportion of total costs allocated to direct and indirect costs was based on estimates from Dall et al. (2010) [37].
Type 2 diabetes (adjusted for costs associated with cardiovascular disease complications)	167.7	65.3	233.0	19.2% of direct medical costs [34] and 38.2% of indirect costs [30] estimated to be associated with cardiovascular disease and therefore, subtracted out from the total costs for type 2 diabetes estimated above.
Colorectal cancer	14.4	--	14.4	Modelled estimates of annual medical costs per case for stages of treatment for adults <65 years and ≥65 years associated with colorectal or prostate cancer in 2010 using SEER [31]. Combined estimate for the total adult US population estimated by combining cost data for all age and treatment categories weighted according to the prevalence of adults in each category [31] and the total prevalence of colorectal cancer in 2016 adjusted to reflect the 2018 US adult population [38].
Prostate cancer	4.7	--	4.7	
Parkinson’s disease	10.0	7.9	17.9	Annual average cost from 2010 [32].
Hip fractures	17.6	--	17.6	Costs of osteoporotic hip fractures among privately-insured young adults (18–64 years) and Medicare-insured elderly adults were compared with matched controls with osteoporosis and no fractures [39]. Direct medical costs were calculated; indirect costs (lost work productivity) were available for a subset of working patients (2006 dollars). The number of hip fractures annually in the US was estimated to be approximately 341,000 (based on patients visiting emergency departments) [40].

B: Billions; SEER: Surveillance, Epidemiology, and End Results; Note: Costs presented are based on costs reported in cited sources and inflated to end of year 2018 US dollars.

**Table 4 nutrients-12-00233-t004:** Dairy consumption among adults (20+ years) in the United States (WWEIA, NHANES 2015–2016) and increase required to meet the Dietary Guidelines for Americans (DGA) recommendation of 3 cup-equivalents/day.

Dairy Product	Dairy Intake among Adults in the US		Scenario 1		Scenario 2	
			Increase Required to Meet DGA Recommendation		Increase Required to Meet DGA Recommendation	
	c-eq/day	g/day	c-eq/day	g/day	c-eq/day	g/day
Total dairy *	1.49	246	1.51	249	1.51	249
Total dairy (Men only)	1.71	282	1.29	213	1.29	213
Milk	0.63	155	0.94	231	1.51	369
Milk (Men only)	0.70	172	0.87	214	1.29	316
Cheese	0.73	49	0.62	41	1.51	101
Cheese (Men only)	0.89	2759	0.46	31	1.29	86
Yogurt	0.09	21	0	0	0.4	100

c-eq = cup-equivalents; g = grams; NHANES = National Health and Nutrition Examination Survey; WWEIA = What We Eat In America. * Based on total dairy consumption, which includes milk, cheese, yogurt, and miscellaneous dairy (e.g., whey). The same values based on total dairy consumption were applied in the models for high- and low-fat dairy products. Scenario 1: Mean intakes of milk, cheese, and yogurt were each increased to result in total proportions by type as specified in USDA Food Intake Patterns [2]. In these patterns, total dairy intake of 3 c-eq/day is comprised of 51% fluid milk, 45% cheese, 2.5% yogurt, and 1.5% soy milk (soy milk is not included within the milk intake). Scenario 2: Intake of each type of dairy product was increased assuming the consumption of only that dairy type to meet the 3 c-eq/day recommendation with the exception of yogurt which was increased 100 g/day (~0.4 c-eq/day) which is the level coinciding with current intake among high-end (90th percentile) consumers in the US adult population.

**Table 5 nutrients-12-00233-t005:** Net annual change in health care costs (Billions $) associated with increasing total, high-fat, and low-fat dairy consumption to the Dietary Guidelines for Americans (DGA) recommendation of 3 cup-equivalents/day among adults in the United States.

Health Outcome	Total Dairy (Billions $ (Range))			High-Fat Dairy (Billions $ (Range))			Low-Fat Dairy (Billions $ (Range))		
	Direct	Indirect	Total	Direct	Indirect	Total	Direct	Indirect	Total
Stroke	1.5 (0.8, 2.3)	0.9 (0.5, 1.4)	2.4 (1.3, 3.7)	--	--	--	0.8 (0, 1.9)	0.5 (0, 1.2)	1.3 (0, 3.1)
Hypertension	3.4 (2.1, 4.1)	0.3 (0.2, 0.4)	3.7 (2.3, 4.5)	2.1 (1.4, 4.8)	0.2 (0.1, 0.4)	2.3 (1.5, 5.2)	2.8 (0.7, 4.8)	0.2 (0.1, 0.4)	3 (0.8, 5.2)
Type 2 diabetes	6.3 (2.1, 12.5)	2.4 (0.8, 4.9)	8.7 (2.9, 17.4)	--	--	--	6.3 (0, 12.5)	2.4 (0, 4.9)	8.7 (0, 17.4)
Colorectal cancer ^a^	1.3 (1.1, 1.6)	--	1.3 (1.1, 1.6)	1.6 (0.5, 2.5)	--	1.6 (0.5, 2.5)	1.1 (0, 2.2)	--	1.1 (0, 2.2)
Parkinson’s disease	−1.9 (−3, −0.9)	−1.5 (−2.3, −0.7)	−3.4 (−5.3, −1.6)	--	--	--	--	--	--
Prostate cancer ^a^	−0.2 (−0.3, 0)	--	−0.2 (−0.3, 0)	--	--	--	--	--	--
Total (primary) ^b^	**10.4 (2.8, 19.6)**	**2.1 (−0.8, 6.0)**	**12.5 (2.0, 25.6)**	**3.7 (1.9, 7.3)**	**0.2 (0.1, 0.4)**	**3.9 (2.0, 7.7)**	**11 (0.7, 21.4)**	**3.1 (0.06, 6.5)**	**14.1 (0.8, 27.9)**
Total (secondary) ^b,c^	**12.5 (6.1, 20.5)**	**3.6 (1.5, 6.7)**	**16.1 (7.6, 27.2)**	**3.7 (1.9, 7.3)**	**0.2 (0.1, 0.4)**	**3.9 (2.0, 7.7)**	**11 (0.7, 21.4)**	**3.1 (0.06, 6.5)**	**14.1 (0.8, 27.9)**

Note: Negative values reflect increased costs; positive values reflect cost savings. Net annual changes in cost within a health outcome across the three dairy types (i.e., milk, cheese, and yogurt) cannot be summed to estimate the net annual changes in costs for total dairy given that risk estimates for each dairy type as well as total dairy are summary measures from different individual studies. Range calculated by summing the lower and upper range of costs and savings from each health outcome. ^a^ Limited to direct costs only, ^b^ Totals reflect rounded sums of unrounded data; ^c^ Includes costs from beneficial outcomes only.

**Table 6 nutrients-12-00233-t006:** Net annual change in health care costs (Billions $) associated with increasing milk, cheese, and yogurt consumption to meet the Dietary Guidelines for Americans (DGA) recommendation of 3 cup-equivalents/day among adults in the United States.

	Milk (Billions $ (Range))			Cheese (Billions $ (Range))			Yogurt (Billions $ (Range))		
Health Outcome	Direct	Indirect	Total	Direct	Indirect	Total	Direct	Indirect	Total
	Scenario 1: Mean Intakes of Milk, Cheese and Yogurt Were Each Increased to Result in Total Proportions by Type as Specified in USDA Food Intake Patterns [2]								
Stroke	2.4 (0.7, 4.2)	1.5 (0.4, 2.6)	3.9 (1.1, 6.8)	--	--	--	--	--	--
Hypertension	2.6 (1.3, 3.8)	0.2 (0.1, 0.3)	2.8 (1.4, 4.1)	--	--	--	--	--	--
Type 2 diabetes	--	--	--	--	--	--	--	--	--
Hip Fractures ^a^	--	--	--	1.7 (1.2, 2.1)	--	1.7 (1.2, 2.1)	--	--	--
Colorectal cancer ^b^	1 (0.7, 1.3)	--	1 (0.7, 1.3)	--	--	--	--	--	--
Parkinson’s disease	−2 (−3.5, −0.7)	−1.5 (−2.7, −0.5)	−3.5 (−6.2, −1.2)	--	--	--	--	--	--
Prostate cancer ^b^	−0.1 (−0.3, 0)	--	−0.1 (−0.3, 0)	−0.3 (−0.5, 0.09)	--	−0.3 (−0.5, 0.09)	--	--	--
Total (primary) ^c^	**3.9 (−1.1, 8.6)**	**0.2 (−2.2, 2.4)**	**4.1 (−3.3, 11)**	**1.4 (0.7, 2.0)**	--	**1.4 (0.7, 2.0)**	--	--	--
Total (secondary) ^c,d^	**6 (2.7, 9.3)**	**1.7 (0.5, 2.9)**	**7.7 (3.2, 12.2)**	**1.7 (1.2, 2.1)**	--	**1.7 (1.2, 2.1)**	--	--	--
**Scenario 2: Mean Intake of Each Type of Dairy Product Was Increased** **Assuming the Consumption of Only That Dairy Type to Meet the 3 C-Eq/Day Recommendation**									
Stroke	3.9 (1.1, 6.7)	2.4 (0.7, 4.2)	6.3 (1.8, 10.9)	--	--	--	--	--	--
Hypertension	4.1 (2, 6.1)	0.4 (0.2, 0.6)	4.5 (2.2, 6.7)	--	--	--	--	--	--
Type 2 diabetes	--	--	--	--	--	--	20.2 (10.1, 33.7)	7.9 (3.9, 13.1)	28.1 (14, 46.8)
Hip Fractures ^a^	--	--	--	4.2 (3.0, 5.2)	--	4.2 (3.0, 5.2)	4.4 (2.5, 6)	--	4.4 (2.5, 6)
Colorectal cancer ^b^	1.6 (1.1, 2.1)	--	1.6 (1.1, 2.1)	--	--	--	--	--	--
Parkinson’s disease	−3.1 (−5.6, −1.1)	−2.5 (−4.4, −0.9)	−5.6 (−10.0, −2.0)	--	--	--	--	--	--
Prostate cancer ^b^	−0.2 (−0.4, 0)	--	−0.2 (−0.4, 0)	−0.8 (−1.4, −0.2)	--	−0.8 (−1.4, −0.2)	--	--	--
Total (primary) ^c^	**6.3 (−1.8, 13.8)**	**0.3 (−3.5, 3.9)**	**6.6 (−5.3, 17.7)**	**3.4 (1.4, 5.0)**	--	**3.4 (1.6, 5.0)**	**24.6 (12.6, 39.7)**	**7.9 (3.9, 13.1)**	**32.5 (16.5, 52.8)**
Total (secondary) ^c,d^	**9.6 (4.2, 14.9)**	**2.8 (0.9, 4.8)**	**12.4 (5.1, 19.7)**	**4.2 (3.0, 5.2)**	--	**4.2 (3.0, 5.2)**	**24.6 (12.6, 39.7)**	**7.9 (3.9, 13.1)**	**32.5 (16.5, 52.8)**

Note: Range calculated by summing the lower and upper range of costs and savings from each health outcome. Negative values reflect increased costs; positive values reflect cost savings. Net annual changes in cost within a health outcome across the three dairy types (i.e., milk, cheese, and yogurt) cannot be summed to estimate the net annual changes in costs for total dairy given that risk estimates for each dairy type as well as total dairy are summary measures from different individual studies. ^a^ Cost data do not allow for distinction between direct and indirect costs; ^b^ Limited to direct costs only; ^c^ Totals reflect rounded sums of unrounded data; ^d^ Includes costs from beneficial outcomes only.

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
