# Peer review of "Health Care Costs and Savings Associated with Increased Dairy Consumption among Adults in the United States"

_nutrients, 2020, doi:10.3390/nu12010233_

Round 1

Reviewer 1 Report

The purpose of the manuscript was to estimate the impact on healthcare cost if adults in the US met the DGA recommendations for dairy consumption. The manuscript is well written,

Major comments:

To meet the DGA recommendations for dairy consumption, it is estimated that Americans adults would, on average, need to double their dairy consumption. If dairy intake increases without adjusting energy intake, this would lead to a substantial increase energy intake, leading to obesity and other co-morbidities. If the diet is adjusted to maintain energy intake, this would lead to  replacement of other foods in the diet that may have different (better) associations with disease outcomes (vegetables and fruits). The AU briefly discuss that in Line 410-420. However, the fact remains that without accounting for the health care costs of displacement of other foods by dairy or the added burden of excess calories intake with greater consumption of dairy, it is hard to see how this modeling exercise can help understand the economic implication of increased dairy consumption, as concluded by the AU. It is not realistic. This is a serious limitation of the study that cannot be explained away in discussion. The justification for the adverse effects (Parkinson's and prostate cancer) being included in the sensitivity analysis and not the primary analysis is questionable. If all diseases associated with dairy intake were found to be of medium or high quality by the criteria used to assess the studies, then they should all be included in the primary analyses. Not doing so inflates the potential healthcare cost saving of greater dairy intake. It is telling that information regarding the sensitivity analysis (or the modeled increase in healthcare cost due to Parkinson's and prostate) was not included in the abstract. The fact that only those that conditions that had adverse associations where excluded from the main analyses and not included in the abstract could be interpreted as biased. The model assumes the effect are linear. The limitation of this approach is discussed in Line 395-409. However, the AU suggest that by selecting for dose-response meta-analyses and present the RR are per dose of dairy strengthened  the analyses. This is not so. For example, the study of Gijsbers [Ref 7] demonstrates that the inverse association plateaus at about 75 g/d, invalidating this assumption. When RR from multiple meta-analyses were available, it is unclear why only one was chosen and the criteria used to do so. Line 197-199, it is unclear whether the values refer to population-wide usual intakes, which is the most appropriate values in this case, or averages of n=5,017 surveyed individuals.

Minor comments:

Line 16, WWEIA abbreviation not used in abstract

Line 21, define c-eq/day

Line 33-35, 3 c-eq for who? All ages? Maybe add this to line 40 to clarify.

Line 43, prevention instead of development? Or both.

Introduction, flesh out both sides, not just benefits of dairy on health outcomes (as a lot of research on adverse effects).

Line 76, the use of 1) is confusing

Line 92, unclear what AU mean by "at risk of chronic disease"

Line 101-103, were these studies excluded? Please clarify.

Page 5, convention to summarize all studies?

Line 153, ; (semi-colon) instead of ) (bracket).

Table 2, statistically significant risk estimates but 0.95-0.99 is a very small reduction, perhaps comment on this  discussion.

Page 13, clarify how low and high fat dairy were assessed and how they fall into scenarios 1 and 2.

Line 320, change indicate to indicates and also, connect this sentence better.

Statement in line 331 is somewhat subjective, why would one assume increase in dairy to be reasonable? If it was, why aren't Americans consuming recommendations already?

Line 382, related to.

In the discussion, perhaps talk more generally about role of dairy products in prevention/development of chronic diseases, specifically the ones included in your analysis.

Line 462, inconsistent instead of consistent?

Table 3, Define SEER

Reviewer 2 Report

This study provided interesting data on the estimated impact on healthcare costs if American adults were to increase their dairy intake to comply with the dietary guidelines. However, I have several questions that must be clarified.

Lines 33-35: Since some readers may be not familiar with the nutrients keys in dairy products please add this information in sentence. Lines 39-40: Since the study was in adults, please refer the prevalence of adults Americans that meet the recommendation for dairy consumption. Lines 76-77: Please add the reference for “published data”. Lines 91-92: According the following inclusion criterion “meta-analysis published in the past ten years of prospective cohort studies”, I don’t understand Figure 2, since literature research were done from 2006 to 2016 and further in 2019. Please explain. Figure 2: Please add information for reasons to exclude after full-text review. The meta-analysis included in this when they refers total dairy products is the sum of milk, cheese and yogurt? Is important to discuss this point, since heterogeneity and inconsistency between studies are may due to, apart from other factors, to food included in total dairy products. May be when you will include information on excluded article after review I understand why you did not identify meta-analysis for obesity, however please explain why, for example, these following meta-analysis were not included? Schwingshackl L, Hoffmann G, Schwedhelm C, Kalle-Uhlmann T, Missbach B, Knüppel S, et al. (2016) Consumption of Dairy Products in Relation to Changes in Anthropometric Variables in Adult Populations: A Systematic Review and Meta-Analysis of Cohort Studies. PLoS ONE 11(6): e0157461. https://doi.org/10.1371/journal.pone.0157461 Sabrina Schlesinger, Manuela Neuenschwander, Carolina Schwedhelm, Georg Hoffmann, Angela Bechthold, Heiner Boeing, Lukas Schwingshackl, Food Groups and Risk of Overweight, Obesity, and Weight Gain: A Systematic Review and Dose-Response Meta-Analysis of Prospective Studies, Advances in Nutrition, Volume 10, Issue 2, March 2019, Pages 205–218, https://doi.org/10.1093/advances/nmy092

Round 2

Reviewer 1 Report

The reviewer appreciates the effort the authors have put into addressing the concerns raised and has no further comments.

Reviewer 2 Report

The authors have satisfactorily amended the manuscript according to the reviewers’ original concerns. 

I thank you for addressing the suggested changes.